# Application of TestBed 4.0 Technology within the Implementation of Industry 4.0 in Teaching Methods of Industrial Engineering as Well as Industrial Practice

**Marek Kliment \*, Miriam Pekarcikova [ID], Peter Trebuna and Martin Trebuna**

Institute of Management, Industrial and Digital Engineering, Faculty of Mechanical Engineering,
Technical University of Kosice, Park Komenského 9, 042 00 Kosice, Slovakia; miriam.pekarcikova@tuke.sk (M.P.);
peter.trebuna@tuke.sk (P.T.); martin.trebuna@tuke.sk (M.T.)
**\*** Correspondence: marek.kliment@tuke.sk; Tel.: +421-55-602-3243

**Abstract:** The paper is focused on the area of the constantly evolving industrial revolution both at the level of individual local economic opportunities and on a global scale. One of the tools of this time called Industry 4.0 is the TestBed 4.0 technology, which helps in its development and opens up opportunities for its use both in research and in practice in manufacturing companies. The paper describes the possibilities and capabilities of the laboratory, which was established as the first of its kind in the Slovak Republic on the campus of the Technical University in Košice in order to combine research activities with practical use directly in the production business processes. Its role is also to involve students in these processes, who, based on the acquired knowledge and experience, will strengthen their position in the labour market after completing their studies. The paper includes a description of available technologies, as well as several case studies carried out at the workplace where this laboratory is located, mainly in the field of industrial and digital engineering.

**Keywords:** TestBed; Industry 4.0; digitization; digital twin

## 1. Introduction

Industry 4.0 is a revolution in the development of production possibilities and technology of today and its near future. Philosophy that will significantly affect not only industry and production itself but the whole direction of society and causes significant changes in it. These changes will involve the way industry is viewed, product development, but will also revolutionize education and employment in the global labour market [1]. TestBed 4.0 is a technology that can be used to demonstrate the benefits of this new era in the development of the industry today and in the future. TestBed is an experimental workplace with state-of-the-art technologies and machinery, which are assembled into a test production line or entire production units. Through them, it is possible to verify new technologies, products, systems and concepts of digitization and digital transformation. They are also used for research, development and innovation projects, and often for educational and demonstration purposes. Such a solution saves time, money and the environment because the operation of the production plant is verified in a virtual environment, does not burn energy, fuel and does not emit emissions. This technology provides a large number of possibilities that can help either in the design of new production facilities and services or in the optimization of existing ones, without a need for large investments in other technologies [2]. At such workplaces, it is possible to practically verify innovative ideas and the application of technology in a digital environment in direct connection with real production. In the conditions of the Slovak economy and industry, such technology is not common at all. Very few companies operating in the Slovak economy use technologies such as modeling of production processes, or their simulation and verification of their production processes using a virtual environment. Such technologies can be found in companies that operate in

the domestic market but are subsidiaries of large national automotive concerns. Small and medium-sized enterprises often do not even accost us with such technologies, because they consider those technologies to be very expensive and do not know how to apply them. The idea to clearly explain to companies what the essence of Industry 4.0 is and what principles it works on, also to permit testing the solutions before implementation in practice and move in the process of digital transformation, laid the foundations for establishing the first TestBed focused on Industry 4.0 processes in Slovakia [3].

This technology can help small and medium-sized enterprises in particular to gain a better foothold on the market. Such types of companies cannot afford to make the wrong strategic decision in fierce competition, as this could cost them their existence, especially in the current economic uncertainty caused by pandemics. The TesBed workplace serves precisely to verify strategic decisions, whether in the area of innovation of production processes and their real impacts, where such decisions can be made without a physical impact on the ongoing production. This means that the company is not forced to stop or interrupt production for it wants to make a change and if that change is not effective, it may lose a lot of time and resources for further change that can help the process. Each change can be tested on a digital model into which the real situation and the current state of the process are transferred. This process is subject to possible changes in the digital environment, which will also be tested in the digital environment and prove their validity. Subsequently, these changes are transferred from the digital to the real process [4].

The location of the first TestBed in Slovakia is of great importance and potential for the future. It is located on the premises of the Technical University in Košice and, in addition to solving orders in the digital industry, students will also be able to become acquainted with its possibilities directly in the teaching process. This fact gives a precondition to the students. These who will use this opportunity should be better prepared for use in real operations in the future, where the field of digitization will be a common part of their work. However, becoming acquainted with such possibilities and technologies opens up great opportunities for them and brings them a great advantage even today on the labour market, where foreign corporations and carmakers operating on the domestic market have a shortage of experts in this field and thus import them to Slovakia from abroad or from their parent companies. People who have the necessary training in process of digitization can become invaluable to such companies on the market. [5].

## 2. TestBed 4.0 and Its Areas of Application

Within the operation and the possibility of using the first Slovak TestBed 4.0, it is possible to research and develop the following areas of the intelligent digital industry at this workplace (Figure 1). It can be said that all areas can be interconnected and applied to each other.

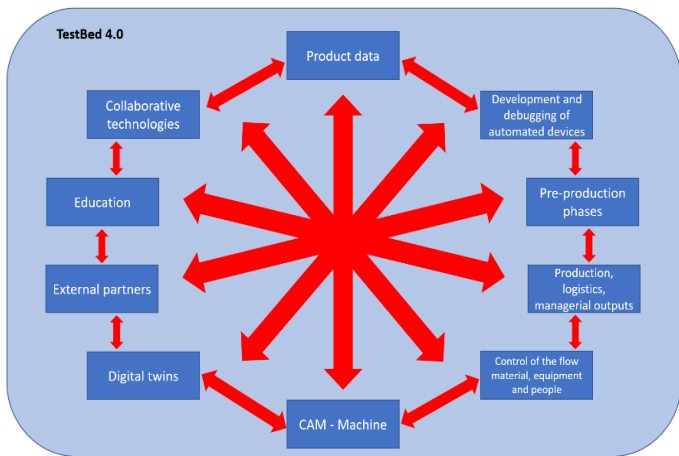

**Figure 1.** Areas of focus of the first Slovak TestBed 4.0.

*2.1. Product Data*

The philosophy of digital enterprises within Industry 4.0 is based on the management of product life cycle data. This data is collected and managed for each product from the first idea of product creation through all its pre-production and production development phases, as well as through marketing, possible modifications to the end of the product life cycle for its recycling. Various information systems and software modules are used to manage this data [6]. An important role in this area is played by the PDM system Teamcenter 13.0, which covers the function of mapping and data collection in all phases of product life and can provide cross-sectional information in its hierarchy for the entire organization and its departments. As part of the application of technology for laboratory equipment, it was necessary to take into account that a company that is a provider of software support for production on the Slovak market within the Siemens consortium participates in its construction and also its use. Therefore, most software is oriented within the scope of this portfolio. However, it should be noted that the use of these technologies in laboratory conditions as well as for teaching students to work in them has great meaning and potential, as these products are used by a large number of companies in the field of industrial and digital engineering. Teamcenter is a commercial platform, but its advantages and functionality should be noted both in the product value chain and in the educational process of students in fields related to industrial engineering. It can display data to users immediately after it has been modified and possibly changed anywhere in the world, without the need to have additional applications installed. It can display drawings and models without having to have any CAx systems installed, it can project the processing and results of simulation models without having to have simulation software. This tool serves both the company's management and all its departments, whether manufacturing or non-manufacturing, to communicate, collect data and solve any problems that may arise during the product life cycle [7].

See software application architecture. Figure 2. Such a platform and its application can also be used in the creation of scientific teams from various fields within the university. The Faculty of Mechanical Engineering of the Technical University in Košice deals with several areas of research and development of various technologies. Such development requires designers, mechanical engineers, mechatronics, electrical engineers, industrial engineers and various other fields. The application of such software is of great importance in the current period during the pandemic. During this period, most professionals have a home office and meetings within the solution teams which are limited to online communication. Data management software can be just a means of this online communication without the need to group entire teams at one time in one place. Each member of the team performs their work, which is automatically stored in the data management system, and each member of the team affected by this change sees it immediately after its implementation and can use it in further work. In this way, a large number of online video call conversations are eliminated. The use of such an application can significantly reduce time for processing research and development tasks, both in the manufacturing company and in the university environment in solving various projects, patents or utility models and developing prototypes of various devices.

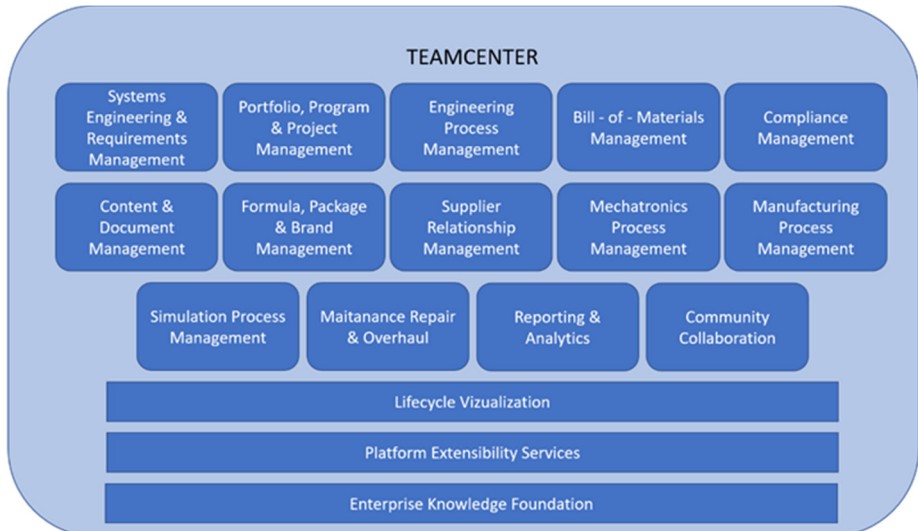

**Figure 2.** Teamcenter and its coverage.

### 2.2. Development and Debugging of Automated Devices

In connection with the data management system and the product life cycle, the development phases of the product are also carried out, after which the possible production methods on automated devices are debugged. After processing the models of individual components, the CAM systems are debugged and generate codes for production on CNC machine tools. For optimal tuning of CNC codes, the programmer or the designer himself must know the machines on which the production will be carried out, he must know the tools that will be used in the production and must also know their properties. It is also necessary to know the properties of the processed materials. After gathering all the necessary informations and processing the model of the part, it is possible to generate the coordinate codes for the movements of the machine tool spindle, as well as for the movements of the tables on which the workpiece is clamped, depending on how many axes the machine tool has and what type of machining goes [4]. The coordinate codes generated at this point are linked to a product lifecycle data management system and are also available at other laboratory sites at the CAM simulation device. They are also part of the pre-production phases of the product but are already directly related to the production itself. CAx software and its modules for various technologies are most often used to process this data and transform it into the correct programming language for the given machine tools. This software must be able to save its outputs in a format with which other software within this technology can then work. [5].

### 2.3. Pre-Production Phases

The pre-production phases include defining the activities on the project, developing a schedule of feasibility and duration of individual activities. Product lifecycle management begins with the preparation of a study and project. Subsequently, in the systems for construction and design, product designs are processed, their functional, resp. component specifications. Construction items are created, which are connected into multilevel structures and logical units. Streamlining product development processes are possible by connecting the PDM systems with CAD systems, the use of integrated tools for document management, workflow and change management. Within this phase, the processing of product models, their decomposition into individual parts is solved. For the preparation of production phases, it is necessary to process the drawing documentation in detail and to create models of all parts of which the product consists. CAD software is used for these activities. Important integration and interconnection of all generated product data. This ensures the connection of all systems using Teamcenter software as part of the processing of a comprehensive product project in the laboratory conditions of TestBed [6].

### 2.4. Production, Logistics and Managerial Outputs

As already mentioned within the conditions of TesBed 4.0, it is possible to plan and verify in detail the entire course of the production process, not only within the production phases but within the entire value flow of the product, from the initial idea of product creation through all its life stages. Within production, it is possible to process models of production processes in detail, to project them into simulations, in which it is possible to interpret all the necessary components of these processes. It is possible to see the outputs from such productions, it is possible to deal with logistics flows, the area of warehousing and other necessary components within the area of production. For managerial decision-making, these virtual simulation models will provide a wealth of data for the ability to make optimal decisions in all areas of strategic business operations [7]. All this can be used both when planning new operations and when optimizing existing operations. Within the possibilities of TestBed 4.0, it is possible to verify the efficiency of investments in new production equipment or equipment modules in a virtual environment. We can virtually integrate these into the existing network of conveyors and corporate infrastructure and verify the effectiveness of such an investment, practically free of charge, without physical intervention in current process. In this environment, we can take into account the necessary human resources in this process and also suggest improvements in the flow of material across the functioning of the entire organization. Based on the results of the simulation, management may decide whether such an investment is acceptable, or consider several possible solutions resulting from the simulation verifications. Creating virtual models allows you to apply and validate any devices that can in some way help increase the efficiency of processes. Simply, it is possible to select a device from libraries to incorporate it into the process and then monitor how this process will be affected [7–9].

### 2.5. Material Flow Control of Equipment and People

The standard option for real-time data collection from the production process is reading data directly from in-house ERP, MES and SCADA software. For real-time data collection, it is possible to use the RTLS (Real-Time Location System) technological tool, which is part of the Smart Factory concept (Figure 3) [9,10].

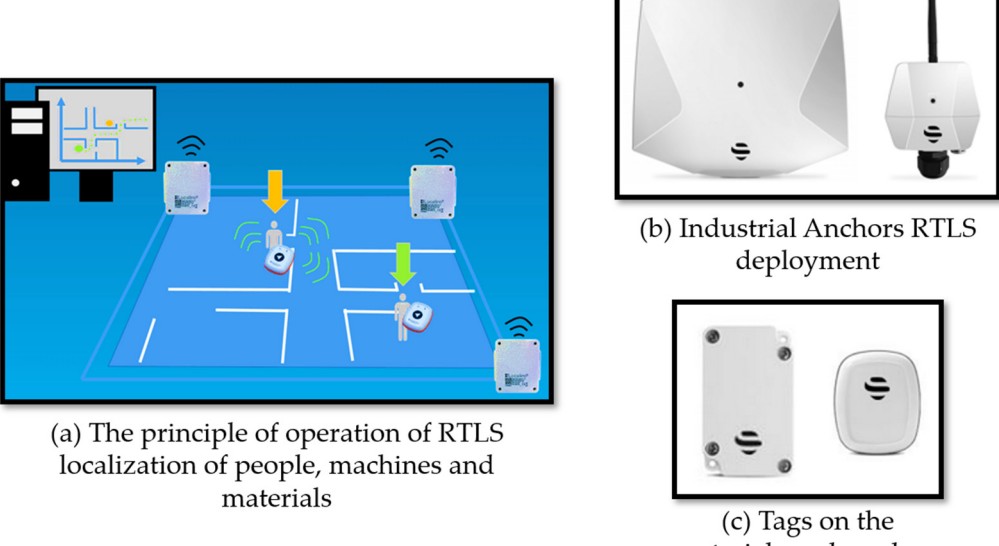

(a) The principle of operation of RTLS localization of people, machines and materials

(b) Industrial Anchors RTLS deployment

(c) Tags on the materials and workers

**Figure 3.** The principle of operation of RTLS localization of people, machines and materials.

Note: schedulability is the ability of tasks to meet all hard deadlines, latency is the worst-case system response time to events, stability in overload means the system meets critical deadlines even if all deadlines cannot be met.

RTLS is a technological solution used for automatic identification resp. monitoring and localization of specific elements of the system with priority in industrial premises in real-time. Such an element can be people, products, machines, carts, crates, pallets, etc., where a given location system can tell managers the location of marked elements in real-time. It is a combination of technology elements (tags, anchors) and software (platform) that gives meaningful location data and transforms it into interactive maps, location tools, thermal maps, dashboards, messages and other functions, depending on the solution provider [11,12].

Basic parameters of the RTLS system:

- Scalability of RTLS localization technology.
- High positioning accuracy—30 cm accuracy that allows full flexibility and variability of virtual zones without any changes of infrastructures.
- Range between anchors in tens of meters.
- Resistance to signal interference in combination with other wireless networks.
- Battery life of mobile tags—several weeks, months to years (depending on the combination of battery capacity, transmission mode and periodicity of identification messages).
- Complete control and configuration of the RTLS system from one application—RTLS Manager.
- Ability to quickly set up and add new devices without the need to reconfigure the deployed system.
- Adjustable localization interval.
- Possibility to modify functionality.
- Possibility of configuration (reprogramming) of individual tags.

The RTLS system based on UWB (Ultra-Wide Band) radio technology uses WiFi networks resp. 4G/5G. RTLS describes a real-time positioning system, a solution that can tell users where a marked system element is located. Several technologies are used to achieve this goal. Some location systems use WiFi, ZigBee, BLE (Bluetooth Low Energy), resp. active RFID. However, not every solution of widely used RFID technology, e.i. radio frequency identification using a microelectronic device (chip and antenna) is possible to realize real-time localization [12]. Table 1 shows some features of localization systems for RTLS.

**Table 1.** Different RTLS technologies comparison.

| Technology | Bluetooth | Zigbee | RFID | UWB |
| --- | --- | --- | --- | --- |
| Accuracy | 10 m | 10 m | 5 m | 10–30 cm |
| Coverage | 10 m | 10 m | 2 m | 100 m |
| Stability | low | low | low | high |
| Security | low | middle | low | high |
| Application | short distance high precision | low accuracy | area entry | high precision |
| Capacity for tag | 50 | 50 | 100 | 500 |
| Anchor quantities requirement per unit area | more | more | more | less |
| Unit price for anchor | low | low | high | high |
| Unit price for tag | high | high | low | high |
| Installation cost | high | high | high | low |

RTLS in combination with Siemens simulation software such as Tecnomatix Process Simulate and Tecnomatix Plant Simulation are one of the important elements of the digital twin. The software stores a log of all movements and interactions in order to facilitate a safe distance, but the same data can be used in decisions about long-term optimization

of production facilities by simulating new production arrangements or workflows to determine which of the options will achieve the required performance. Real-time data processing solutions between RTLS and TX Plant Simulation, resp. TX Process Simulate that is possible in the TestBed environment is focused on [13]:

- Make-to-order (MTO) production approach, focused on fast response and personalization of products.
- Reduce the lead time—the overall lead time is shortened thanks to the absence of waste and gaps in production.
- Use digital work orders—carrying the information and extending it with real-time location and other data types based on the sensors used.
- Higher visibility and increased ability to plan, using heatmaps and spaghetti diagrams in real-time mode.
- eKanban—continuous flows in real-time.
- Optimize fleet overall efficiency (OEE) by revealing inactive periods and fixing them, it is possible to compare the OEE data of each vehicle, etc.

By combining RTLS and simulation technologies from Siemens, it is possible to quickly and efficiently model interactions within material and logistics flows, the movement of employees located on the production line. The potential can be seen in the redesign of production lines, in solving persistent safety problems, in reconfiguring workers to increase productivity [13].

By combining software from Siemens Digital Industries Software with the technology provided by Sewio, it is possible to map logistics and material flows in production very efficiently with the technology provided by Sewio, it is possible to map logistics and material flows in production very efficiently. Model situations of material movement of workers or other elements in the production process can be practically simulated and tested directly in the laboratory. Potential customers who would want to apply this technology in their operations can demonstrate its functionality directly in the laboratory, which is equipped with RTLS sensors in every corner of the laboratory. These sensors read and record into the system the movement of tags and controls, which can be attached to the material or given to a worker who will move in space. Another possibility of using material flow mapping is to use RFID sensors, which are located on individual tables and laboratory stations, and by reading tags in individual locations, it is also possible to record the movement of products or employees, which use sensors to record every production operation. In this way, snapshots of work activities can be obtained [14].

### 2.6. CAM-Machine

A very interesting area within the verification of functional properties and correctness of the computer version of the product model and subsequent generation of code for the machining process is its connection to a real device located in the workplace. It must be said that this is not a real CNC machine of real dimensions, which would cost thousands of euros, but a modular model. From this model, we can assemble a machining device similar to reality within the possibility of using several machining axes, several machining tools and similar assets. Most importantly, however, such a device can realistically verify and produce the proposed model, which is many times a miniature of the actual part but it is sufficient to physically verify the correctness of the programming code for CNC devices. The designer in CAD software creates a model of the part. The programmer transfers this part to the CAM software environment and defines the machining properties of a specific material and equipment on which the machining will be performed. Subsequently, the CAM software connects the model and the generated code to a modular device which, such as a functional machine, can move in several axes and machines with several tools [15]. The modular device replacing the real machine is set in motion and the required produced part is based on the CAM software-defined code (Figure 4).

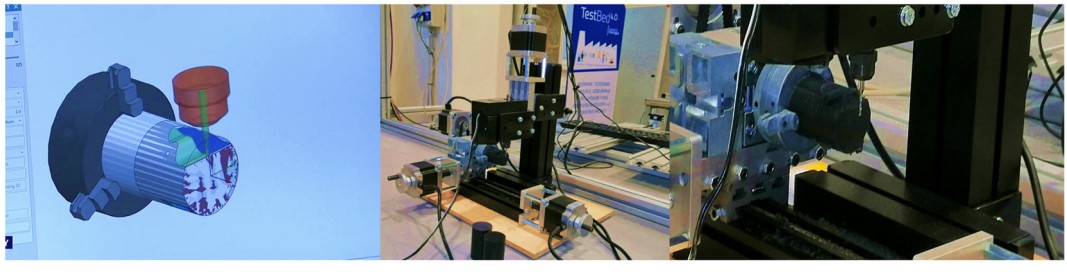

| (a) Simulation of part machining in CAM software | (b) Functional modular model of CNC machining center | (c) Detail of workpiece machining |

**Figure 4.** Verification of CAM design in real conditions.

### 2.7. Digital Twins

Time is a very strong factor in the supply chain. Both the development time and the delivery time of a device or product to the customer. If a company embarks on the necessary path of a process automation today, it will encounter a problem with the supplier of such technology, as there are not many such suppliers on the market, this prolongs the delivery time of a required equipment. It should be noted that each operation that plans to apply some elements of automation is individual and what suits one customer will not satisfy requirements of another customer. In such an application, it is necessary to adapt a device to the operation in which it will subsequently be used. The staff and equipment of TestBed 4.0 can offer assistance to both equipment manufacturers (integrators) and automation procurers. Using MCD (Mechatronic Concept Design) technology, it is possible to simultaneously develop a workplace design and program the device controls. This synergy can reduce development time and thus delivery of the device to the end-user [16,17].

The workplace is equipped with technology that, in addition to the design of the device itself, can simultaneously solve its functional programming with it. The device is created in a virtual environment (Figure 5a) and it can be anything, a CNC machine, a parts feeder, a manipulator and similar assets. The virtual environment is connected using PLC systems (Figure 5c) to the actual control panel of the device, which can be used to test the programming itself and use in a real operation (Figure 5b). In this way, it is possible to teach employees to operate the equipment before the equipment appears in the operation. Such a connection of the virtual model with the actual controls can also be defined as a digital twin of the device [18].

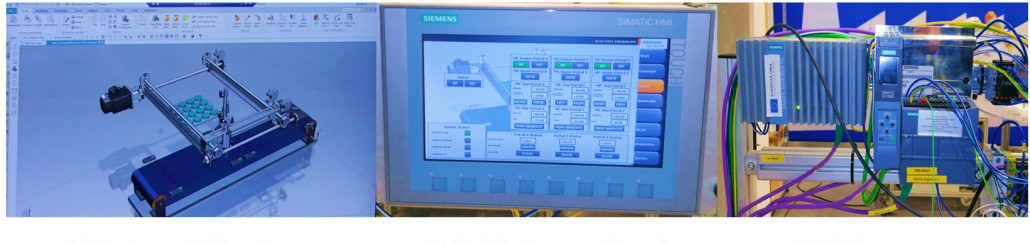

| (a) Device model in software | (b) Real device control panel | (c) PLC converter |

**Figure 5.** Virtual machine control system in TestBed 4.0 conditions.

As already mentioned, the workplace has the ability to create a digital twin in human-machine interaction. On other stations, however, it can create other types of interactions of the virtual world with the real one. In the workplace, it is possible to test a digital twin in several directions. It is possible to control the operation of physical devices by simulation with the help of transmitted signals. It is also possible to control the simulation parts using signal transmission and thus to verify possible states in the production process [19,20]. The workplace offers a practically unlimited possibility of connecting sensory elements and with the help of them transferring real states in real operations to virtual models. Based on

the use of sensors, there is a development in the field of digital twins, which will continue to develop and move to an ever-higher level within digital and intelligent companies. The philosophy of the digital industry builds its mainstays on this principle and allows workplaces such as TestBed to move further within their application [21].

The main factor in the interaction of simulation with the real environment is sensors. This would not work if the signal transmission between it and the virtual environment did not work. This signal transmission is ensured using PLC converters, which connect these two worlds, and with this help, it is possible to achieve the required signal transmission both from the simulation to the real world and in the opposite direction within the control and programming. With the help of such devices, it is possible to collect an incredible amount of data from devices and operations, and this data can later be used in the field of virtual environments and solutions for optimizing these devices [22]. The average operator and user of a given device often have no idea what can be monitored, captured and then used to improve the efficiency of a given device or process. Within the workplace, two types of PLC converters are used, namely, PLC units S7-1500 and S7-1200, to which various types of industrial sensors are connected (Figure 6). These units can be programmed as needed and connected to the required type of sensor or controllers. The Simatic programming language is used for programming [23,24].

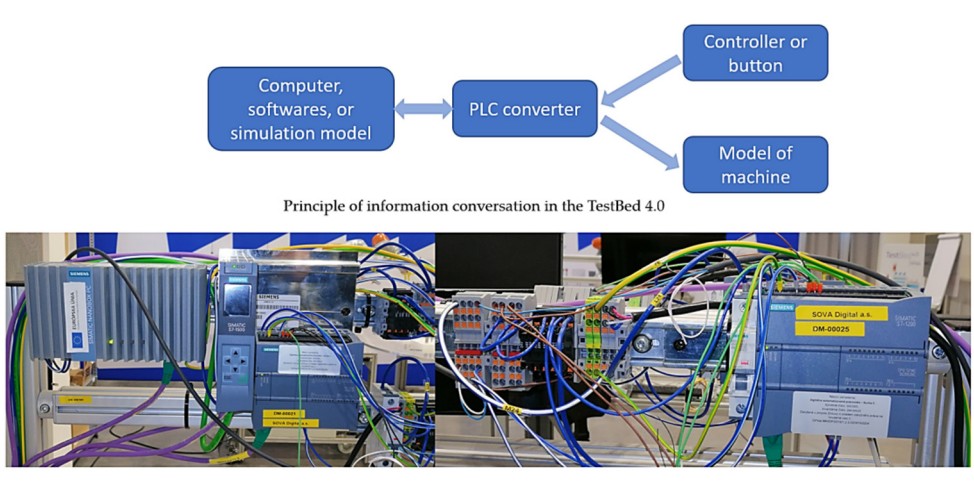

**Figure 6.** PLC converters.

The most widely used software in the field of creating digital twins in the form of simulations is Tecnomatix Plant Simulation. In this software, virtual images of real productions are created, which are either connected by PLC and sensors to the real environment and with the help of these sensors parts of the simulation are controlled, or the simulation itself sends signals using devices to demonstrate the operation of individual workplaces. Figure 7a demonstrates the line operated by workers. With the help of sensors and controls Figure 7b the individual operators operating the equipment are put into operation. The simulation evaluates the results and occupancy of the workplace according to the current state and activity of operators who are either active or inactive thanks to sensor control [25].

Another example is the control of a robotic workplace (Figure 8). The robot is in a security cage. Using a sensor that can be either activated or deactivated with a simple sliding movement, we control the door on the cage of the robot, which works when the cage is closed, if the door is open, it stops immediately. Instead of a sensor, it is possible to connect, for example, a photocell or a light gate, which automatically deactivate the robotic workplace when motion is detected. In this way, it is possible to test safety elements in production [26].

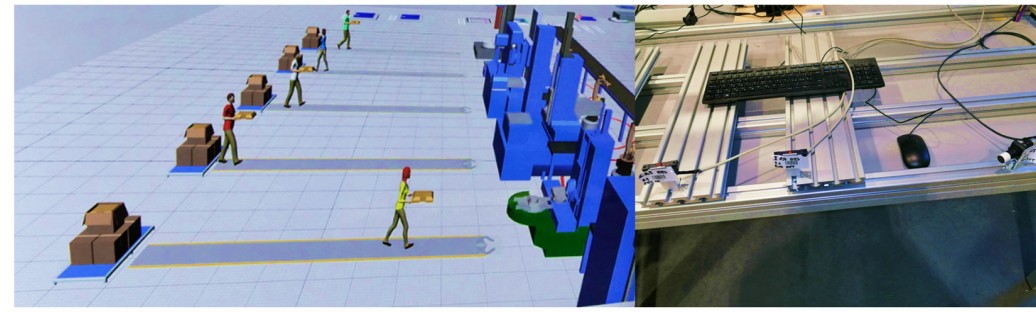

(a) Operation of various workplaces in a simulation environment

(b) Sensors and controllers controlling various operator workstations in simulation

**Figure 7.** Sensor-controlled workplaces for operating individual devices.

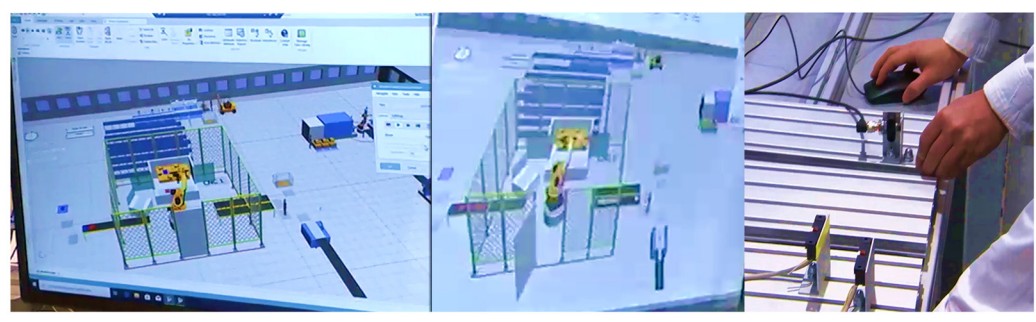

(a) Robot cage closed  (b) An open robot cage  (c) Robot cage door controller

**Figure 8.** Control of a virtual robotic workplace with the help of sensors.

The opposite example of the use of digital twin technology, where the simulation transmits signals and thus puts a real device into an operation, is a simulation of a workplace operated by an operator. In the simulation, the operator brings a blank to a lathe (Figure 9a). Once the blank is stored in the machine, the simulation sends a signal, which is transmitted by a PLC to a real modular model of the lathe on the table in operation and this is activated by spinning the chuck of the device (Figure 9b). Once the operation is completed in the simulation, the chuck stops and set in motion when the operator brings another blank to the machine. In this way, it is also possible to verify the energy load of individual devices in a real operation [27,28].

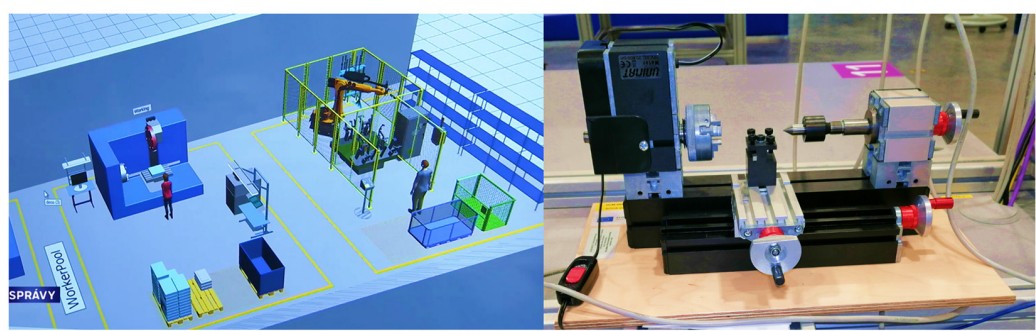

(a) Simulation model of a turning workplace

(b) Modular model of a lathe in operation

**Figure 9.** Simulation controlled device.

These and many other ways of application and testing are offered by devices oriented to the area of digital twins in the TestBed 4.0 workplace [29].

Using simulation in the Tecnomatix Plant Simulation software module, several simulation models were developed to improve the efficiency of production processes. All these

simulations can be connected to the TestBed technology and using this technology to verify various states in the production process. Initially, the simulation addressed the justification of the use of robots in product packaging. In this case, it is a production of cod and lettuce in a food plant, several bottlenecks were addressed in the production, but the main task was to verify what increase in production will occur when replacing manual packaging by workers with an automatic robotic workplace. The original condition can be seen in Figure 10. Here we can see that in 8 h of work shifts, workers manually wrap 3 pallets with salad and 4 pallets with cod. There are 384 packages of both salad and cod on one pallet. One package contains 12 boxes of products, which means that there are 4608 boxes of products on one pallet.

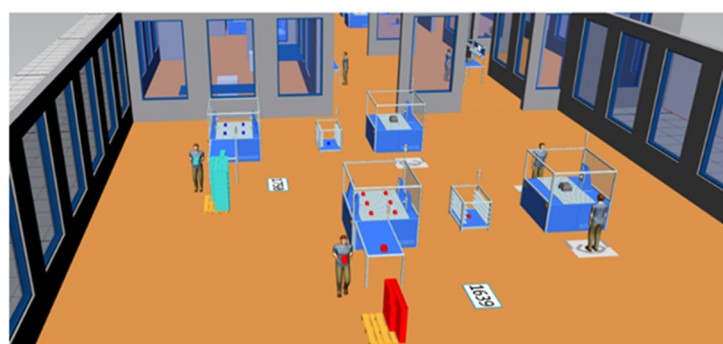

| Cumulated Statistics of the Parts which the Drain Deleted | | | | | | | | |
|---|---|---|---|---|---|---|---|---|
| Object | Name | Mean Life Time | Throughput | TPH | Production | Transport | Storage | Value added | Portion |
| Prve_poschodie.Drain | Pallete_of_salad | 3:30:43.7010 | 3 | 0 | 40.62% | 59.38% | 0.00% | 0.00% | |
| End_of_production | Pallete_of_cod | 3:24:47.7288 | 4 | 1 | 43.22% | 56.78% | 0.00% | 0.00% | |

**Figure 10.** Original condition in the production plant.

In this production process, the manual packaging workplace was replaced by an automated robotic workplace in the packaging of both types of products. The results of such a product innovation can be seen in Figure 11. In the picture, we can see the situational model in production and its statistics with the improvement of the efficiency of this workplace. We can see that the line in this state packed 6 pallets of salad and 7 pallets of cod. Such a change can increase production by almost 100% in both cases. It can therefore be said that the effectiveness of such an investment is likely to be justified. The company has started to address this issue as it has a high demand for its products and is addressing an increase in production [30,31].

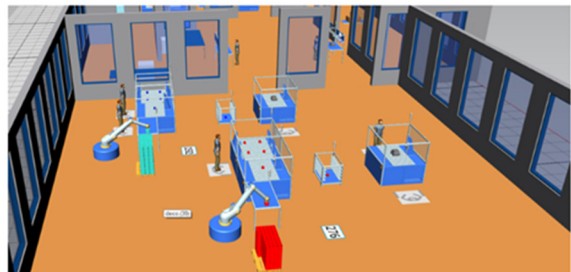

| Cumulated Statistics of the Parts which the Drain Deleted | | | | | | | | |
|---|---|---|---|---|---|---|---|---|
| Object | Name | Mean Life Time | Throughput | TPH | Production | Transport | Storage | Value added | Portion |
| Prve_poschodie.Parizsky_salat | Pallete_of_salad | 2:53:00.1738 | 6 | 1 | 45.53% | 54.47% | 0.00% | 0.00% | |
| Hotovy_produkt | Pallete_of_cod | 2:31:12.5830 | 7 | 1 | 46.15% | 53.85% | 0.00% | 0.00% | |

**Figure 11.** Application of robotic packaging in the production process.

At the TestBed workplace, it is possible to control the individual packaging workplaces by connecting the simulation with sensors on the laboratory control panel and to switch them on and off using sensors, thus simulating possible production faults and monitoring production results without faults and in the event of a fault.

In the next solved operation, the addition of a new equipment to the production process and the increase of its fluidity using the implementation of conveyors in the production process were verified using simulation. The simulation solves the production process of saw blade production. In the original state, the material flow between individual operations is provided by workers operating individual devices. At the beginning of the production process, one laser cutter works in its original state, which cuts the necessary product shapes from the sheet metal. The production process is visible in Figure 12. It can be seen from the simulation output statistics that the plant produces 126 product packages per shift [32].

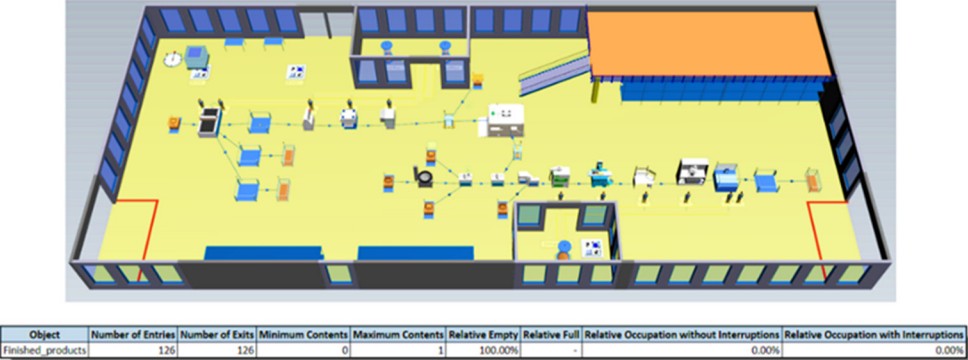

| Object | Number of Entries | Number of Exits | Minimum Contents | Maximum Contents | Relative Empty | Relative Full | Relative Occupation without Interruptions | Relative Occupation with Interruptions |
|---|---|---|---|---|---|---|---|---|
| Finished_products | 126 | 126 | 0 | 1 | 100.00% | - | 0.00% | 0.00% |

**Figure 12.** Production process in its original state.

The improvement of the production process consists of the implementation of conveyors between some production operations. This proposal will reduce the physical load on workers and based on the output of the simulation model, it is visible that the output is also increased by 46 packages of products. Another improvement is the application of another laser cutter at the beginning of the production process. In its original state, the cutter was very busy and it was necessary to increase production. The streamlined production process is visible in Figure 13 [33].

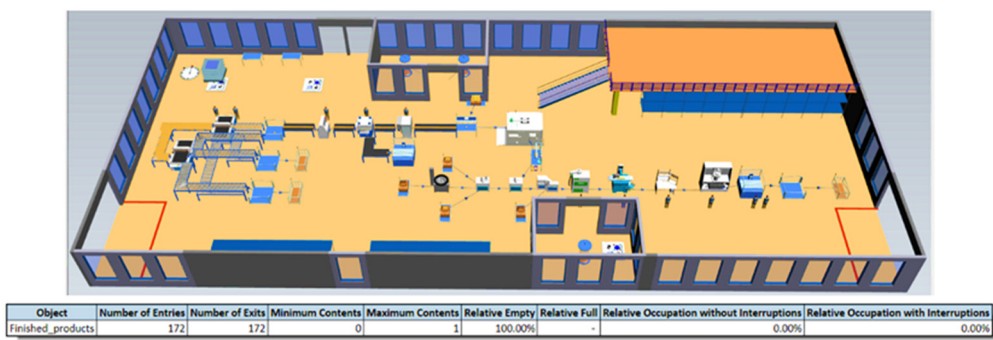

| Object | Number of Entries | Number of Exits | Minimum Contents | Maximum Contents | Relative Empty | Relative Full | Relative Occupation without Interruptions | Relative Occupation with Interruptions |
|---|---|---|---|---|---|---|---|---|
| Finished_products | 172 | 172 | 0 | 1 | 100.00% | - | 0.00% | 0.00% |

**Figure 13.** Streamlined production process using simulation.

### 2.8. External Partners and Education

This specialized workplace is located on the premises of the Faculty of Mechanical Engineering of the Technical University in Kosice on the premises of the Institute of Industrial and Digital Engineering Management and was established in cooperation with these institutions and a private company dedicated to digitization for a long time. The company, which was involved in the construction of this workplace, solves real problems in various areas and from practice, at the same time connects real external partners with practice with educational institutions in this area [29]. At the same time, it plans to involve students and university staff in the process of solving real tasks from practice. There is space for education of young researchers as well as newly trained people, who will

be very much needed for practice in the future. By enabling students to work in such a specialized workplace and to encounter real business problems in practice, students will become invaluable after graduating from the labour market in this emerging area of the new era of Industry 4.0 [34,35].

### 2.9. Collaborative Applications in Robotics

A very important part of the TestBed 4.0 workplace is the area of collaborative robotics. At this workplace, there is a collaborative Kuka robot, which can be involved either in testing and education in the field of programming such robots, or the field of digital twins in the field of robotics. In Figure 14 it is possible to see a collaborative robot performing a simple manipulation of the nuts, which according to the default program moves from one station to another. Such and many other simple and complex operations can be tested by students and researchers within this workplace [36,37].

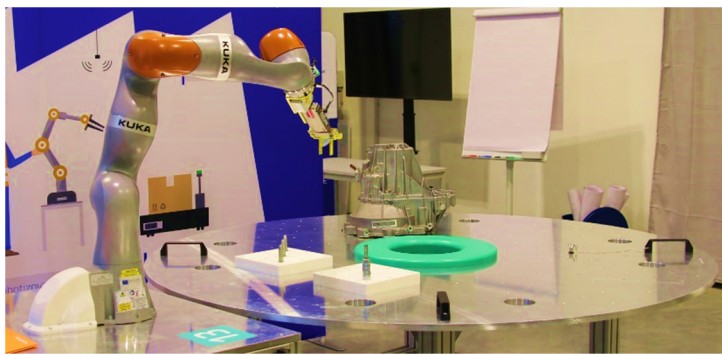

**Figure 14.** Collaborative KUKA robot for easy handling.

Below in Figure 15, it is possible to see a simulation of a robotic workplace, which is created in the Tecnomatix Plant Simulation module, which was run as a demonstration of applying the possibilities of this workplace remotely when opening the TestBed 4.0 workplace in its operation in Kosice. The real robotic workplace was located at the other end of the Slovak Republic. The simulation controlled a real workplace, which performed manipulation tasks in real-time both in the simulation and in a real environment hundreds of kilometres away [38,39].

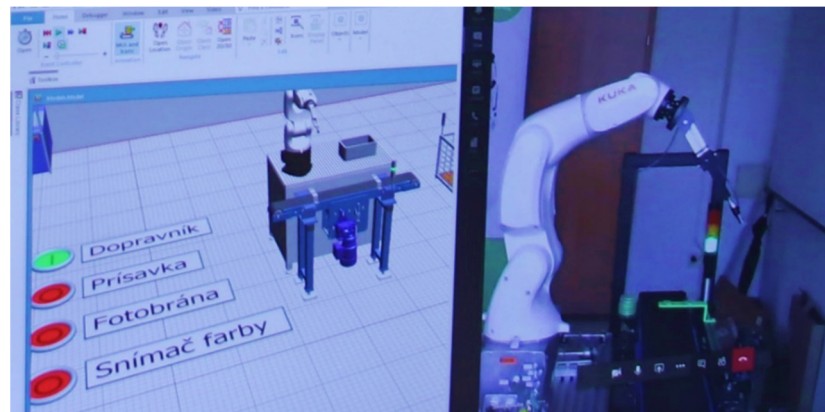

**Figure 15.** Robot control using remote simulation.

### 3. Conclusions

The field of digitization and intelligent business systems is constantly evolving and will continue to evolve. Therefore, it is necessary to keep up with this trend or determine development trends in it. Such possibilities are offered to us by the TestBed 4.0 workplace,

which is directly subject to and demonstrates what can be solved with the help of digitization in the currently evolving era in not only product manufacturing and service delivery, but also in the entire development of society, such as Industry 4.0 is undoubted. The paper describes several possibilities and examples of using such a workplace in improving efficiency in industry, but also in the educational process. This workplace is the first in the Slovak Republic, but after companies know it better and understand its possibilities and capabilities, there is no doubt that such workplaces will be created gradually in various industries. That is why it is necessary to use the location of this workplace on university grounds and prepare as many students as possible for them, who will be invaluable people on the labour market in the future, when digitization and smart industry will be common part of life, especially in the development of the domestic economy. Such workplaces can give companies a huge competitive advantage and can make it easier for them to improve the efficiency of their processes, either in production or in other areas within the value flow and life cycle of the products.

As a part of the research and development of new technologies on university grounds, this workplace offers great potential for the possible development of new technologies or the improvement of existing ones, whether in the field of simulation, logistics, warehousing and similar fields. Thanks to these modern technologies, research in many areas of industrial and digital engineering is also offered in these areas. It will involve ever-deepening research in the field of digital twins, collaborative robot workplaces, material flows and their resources, and many other research and development opportunities. Due to the measures, it has not yet been possible to carry out several research activities that are planned for the future and will be further interpreted and published in the future due to the current situation within the pandemic, which has global consequences. As a part of research tasks and the solution of several projects, several other workplaces are being built at the workplace within which this laboratory is located, which should expand the functionality of the current workplace. The aim is to make it possible to research and solve, as far as scientifically or on a practical scale, as many problems as possible that may arise in production. All existing and existing workplaces under construction within this issue should be networked and could cooperate in solving joint scientific and research tasks, whether within a university or in a corporate field.

**Author Contributions:** Conceptualization, P.T. and M.K.; methodology, M.K.; software, M.P. and M.K.; validation, P.T. and M.T.; formal analysis, P.T.; investigation, M.P. and M.K.; resources, M.K. and P.T.; data curation, M.P. and M.T.; writing—original draft preparation, M.P., P.T., M.K. and M.T.; writing—review and editing, M.P.; visualization, M.K.; supervision, M.T.; project administration, P.T. and M.P.; funding acquisition, P.T. and M.P. All authors have read and agreed to the published version of the manuscript.

**Funding:** This research was funded by of the grant project APVV-17-0258 "Digital engineering elements application in innovation and optimization of production flows", APVV-19-0418 "Intelligent solutions to enhance business innovation capability in the process of transforming them into smart businesses", VEGA 1/0438/20 "Interaction of digital technologies to support software and hardware communication of the advanced production system platform", KEGA 001TUKE-4/2020 "Modernizing Industrial Engineering education to Develop Existing Training Program Skills in a Specialized Laboratory" and KEGA 009TUKE-4/2020 "Transfer of Digitization into Education in the Study Program Business Management and Economics".

**Institutional Review Board Statement:** Not applicable.

**Informed Consent Statement:** Not applicable.

**Data Availability Statement:** Not applicable.

**Conflicts of Interest:** The authors declare no conflict of interest.

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
