# Peer review of "Application of TestBed 4.0 Technology within the Implementation of Industry 4.0 in Teaching Methods of Industrial Engineering as Well as Industrial Practice"

_sustainability, doi:10.3390/su13168963_

Round 1

Reviewer 1 Report

The content of the manuscript corresponds to its subject and is highly topical in today's digitization age and improvement of processes in the context of Industrial 4.0. Description and possibilities of technology TestBed 4.0 is an interesting topic for research in this area.

The abstract briefly describes the specifics of Industry 4.0 and what it brings, it also mentions theTestBed technology, but to a small extent describes the overall goals of this technology and the manuscript itself. I suggest to specify in the abstract more information about the possibilities and goals of the paper.

In the chapter 2.2 is stated: They are also part of the pre-production phases of the product, but they are already directly related to the production itself. NX software and its modules for various areas in Cax technologies are most often used to process this data and transform it into the correct programming language for given machine tools. Is it necessary to state the name of the commercial software at each point? Is it not possible to make such changes in another module, or in general in the Cax system ?

Authors also list several commercial software in Ch apter 2.3. I would generalize these points and not focus on specific software. In my opinion, it is possible to process data in most CAD software.

In Chapter 2.5, is stated that in the combination of software from Siemens and Sewio, it is possible to solve optimization more efficiently, are these 2 software packages compatible and interoperable? The text below Figure 3 is not entirely clear. Is Figure 3 is the author‘s own elaboration, or is just downloaded from the RTLS system provider?

In Chapter 2.7, Figures 5 and 6 show the means for the digital twin, the images are quite vague, I suggest adding a diagram explaining how the data is converted using these technologies.This chapter is quite extensive compared to the others, I appreciate the sample of case studies which authors presented.

In conclusion, in addition to pedagogical and practical goals, I would also highlight the scientific goals and possibilities of such technology.

In the article, I would re-evaluate the mentioned literary sources and supplement them with more current sources of information in the researched area. I would also reconsider the need to cite the commercial software from a particular provider everywhere.

In summarize, the form of the paper is interesting. I propose to the authors to incorporate the submitted proposals and thus improve the overall form of the paper. I also propose to check compliance of your article with the instructions for publication of the paper in the journal. After incorporating the changes, I suggest publishing the article.

Author Response

Hello

Please find attached a modified post based on reviews from all reviewers. I have edited some texts, pictures, added literary references and the like. Contact me in case of any questions.

Thank you very much 

Reviewer 2 Report

Dear authors,

In the attachment you will find the review of your manuscript and the requirements for its improvement before publication.

Best regards,

Author Response

(The authors gave the same response as above.)

Reviewer 3 Report

For such a technical article the level of citation and referencing is sparse. 

As a reviewer, I do not want over citing when it is not necessary BUT there are only 27 citations in such a technical article. There are numerous examples of paragraphs with nothing to back up the statements or commentary. 

Also, the citations are not in numerical order -should be [1,2,3], not [1,3,11 etc etc]

The abstract is not an abstract -please rewrite

The research questions should be clearly articulated at the end of the introduction. 

The conclusion could be stronger -and more specific around originality findings, future opportunities and applicability. 

Author Response

(The authors gave the same response as above.)

Reviewer 4 Report

The article is interesting, as it presents a new technology, Testbed 4.0, to make students familiar with it at university, and by doing so increasing their future skills, making them more valuable on the labor market. In section 2 the authors describe all parts that are presented in figure 1. Maybe it could make things more clear to explain what Testbed 4.0 is all about in the text above this figure. In this way it becomes more clear that Testbed 4.0 consists of a lot of digital features, and it is introduced better, before the different features are elaborated.

Please look after the English, and the references, which are not according to the criteria from Sustainability.

Specific comments

Line 99-106 and Line 107-113 are exactly the same! Both are about ‘It can display data to users immediately after it has been modified and possibly changed anywhere in the world, without the need to have additional applications installed. It can display drawings and models without having to have any CAx system installed, it can project the processing and results of simulation models without having to have simulation software. This tool serves both the company's management and all its departments, whether man-ufacturing or non-manufacturing, to communicate, collect data and solve any problems that may arise during the product life cycle.’

Line 113-114. The authors write ‘See software applica-tion architecture. FIG. 2.’. Where can I find this software application in figure 2, as it is not mentioned explicitly?

Author Response

(The authors gave the same response as above.)

Round 2

Reviewer 3 Report

I am very disappointed by the lackadaisical response to the reviewer's comments -there was no document outlining what changes were made or where, I do not expect the author to document every single change in terms of citations, etc but they could have summarised revised sections that were particularly  called out my myself or other reviewers.  While it does seem a lot of changes were made -the authors could at least answer the comments made originally.    

Author Response

Dear reviewer, thank you for the guidance and advice for improving my contribution. Based on your review as well as the reviews of other reviewers, I made the following adjustments to the post:

- I have modified the abstract and introduction,
- I added some citations from WOS and SCOPUS databases to the article
- I have generalized some descriptions in the text so that I do not have to give specific names of commercial software
- I edited some images and changed their description based on the magazine template.
- In the conclusion i have added few sentences.

Attached to this answer is the latest version of the paper, which has been checked in English and some grammatical errors and words have been corrected compared to the original paper.

Thanks for the review and valuable tips for improving the level of my post.
